## [Peer Review File · EMBO Reports]

Somatic gene repression ensures physical segregation of germline and soma in *Drosophila* embryos

Miho Asaoka, Mizuki Kayama, Tomoki Kawagoe, Makoto Hayashi, Shumpei Morita, and Satoru Kobayashi

Corresponding author(s): Satoru Kobayashi (skob@tara.tsukuba.ac.jp) , Miho Asaoka (masaoka@tara.tsukuba.ac.jp)

Review Timeline:

Transfer Date:	21st Oct 25
Editorial Decision:	13th Nov 25
Revision Received:	19th Dec 25
Editorial Decision:	19th Jan 26
Revision Received:	22nd Jan 26
Accepted:	28th Jan 26

Editor: Achim Breiling

Transaction Report: This manuscript was transferred to EMBO reports following peer review at Review Commons.

**Review
COMMONS**

Review #1

1. Evidence, reproducibility and clarity:

Evidence, reproducibility and clarity (Required)

Asaoka et al. investigate how *Drosophila* PGCs (pole cells) maintain physical segregation from somatic cells during early embryogenesis. While it is well established that the pole form at the posterior of the embryo and remain outside the somatic epithelium until mid-embryogenesis, the mechanisms that enforce this separation are not fully understood. Previous work by the authors and others showed that the translational repressor Nanos and the transcriptional silencer Pgc (Polar granule component) act together as a "double-lock" system to suppress somatic gene expression in pole cells. Nanos inhibits the translation of importin- α 2, a key adaptor for nuclear import, thereby blocking the entry of transcriptional activators into the pole cell nuclei. Pgc, meanwhile, represses transcription by preventing RNA PolII CTD phosphorylation. Here, the authors ask if and how repression of somatic genes prevents pole cells from adopting somatic fate, by analyzing pole cells lacking both components of the double-lock mechanism (embryos from *pgc* mutant females and from females overexpressing importin- α 2 (*Impa2*)).

Transcriptomic analysis revealed significant misexpression of somatic genes, including 389 transcripts uniquely upregulated only when the double-lock mechanism was disrupted. Importantly, genes associated with germline identity were downregulated. Live imaging and morphological analyses showed that, unlike their spherical wild-type counterparts, the mutant pole cells formed abnormal cellular protrusions as early as stage 6 and invaded the adjacent somatic epithelium of the midgut, a region where pole cells are normally excluded. Furthermore, these aberrant pole cells often underwent apoptosis.

To identify the misexpressed somatic genes responsible, the authors focused on genes involved in neuronal morphogenesis, reasoning that similar protrusions are seen in neuroblast lineages. They identified Miranda (*mira*), normally exclusive to somatic cells, as being ectopically activated in mutant germ cells. Strikingly, *mira* knockdown in double-lock mutants rescued the abnormal protrusion and invasion phenotypes, and also reduced cell death, implicating *mira* as a key driver of these defects. Single-cell RNA sequencing of double-lock mutant pole cells revealed that cells at the periphery of the pole cell cluster, which inherit less germ plasm, were more likely to ectopically express *mira*, form protrusions, and invade the epithelium. These findings suggest that both repression of somatic genes and sufficient germ plasm inheritance are essential to safeguard pole cell identity and ensure physical separation from the soma.

In summary, Asaoka et al. provide convincing evidence that silencing of somatic gene

expression programs by Nanos and Pgc is necessary for the germline-soma distinction - perhaps most interestingly, the physical separation of PGCs from the soma. Failure to do so leads to cell identity confusion, physical invasion into the soma, and eventual germ cell loss through apoptosis. Although it may not be surprising that ectopic expression of somatic genes leads to identity confusion, it is surprising that among 389 upregulated genes, *mira* seems to be the main culprit. This part of the manuscript is less well developed and less satisfying - how and why ectopic expression of *mira* would have such a major consequence for PGCs is not considered in any depth.

****Major Concerns:****

1) Figure 2: The authors analyze pole cell protrusions during early embryogenesis, but it remains unclear how these morphological changes in the pole cells impact the end result of gonad formation or germ cell development. Further discussion or experimentation linking these early events to later defects in germline maturation would strengthen the developmental relevance of these findings. Also see point #5 below.

2) Figure 3: The authors suggest that the pole cells forming protrusions are the same ones that later invade the midgut epithelium and undergo apoptosis. However, there is no direct experimental evidence connecting these two observations. While live imaging using Moesin-GFP is limited by the loss of the protrusion phenotype as PGCs migrate, staining for Hid or Atg1, autophagy-related proteins known to induce cleaved Dcp-1, could provide a clearer link. This is especially important given the transcriptomic upregulation of autophagy genes in early pole cells, potentially allowing identification of protruding cells destined for apoptosis.

3) The authors utilize the MatGal4 driver, which drives expression broadly during oogenesis and the embryo (including the soma). This raises the concern that pole cell phenotypes observed with *mira* RNAi could reflect effects from depletion of *mira* in the oocyte or in the somatic cells at the posterior, which could influence pole cell behavior (ZGA in soma precedes ZGA in pole cells). How do the authors distinguish between PGC-specific effects and possible somatic contributions? This issue may also explain why central pole cells appear unaffected, while peripheral pole cells, those in closest proximity to the soma, exhibit defects. Overexpressing *mira* could help to determine whether the phenotype is intrinsic to pole cells or influenced by their somatic microenvironment. If overexpression fails to induce protrusions in central pole cells, this would suggest a non-cell-autonomous role of the soma in driving the observed behavior.

4) While the authors show that *mira* (either in pole cells or soma, see above) is necessary for the protrusion phenotype in peripheral pole cells, the mechanism by which *mira* promotes these morphological changes is unclear. Functional studies examining how *mira* contributes to membrane dynamics, polarity, or cell motility could clarify its role. Additionally, the study focuses solely on expression of *mira* RNA, without addressing *mira* protein localization or function. What does *Mira* protein do (in other contexts) and how might this function relate to a function in pole cells?

5) What happens to the central pole cells in *pgc- impa2OE* embryos? Do they survive and populate the gonad? This should be addressed.

6) Why would *Nanos* and *Pgc* be sufficient to repress *mira* in the peripheral pole cells but not in the central ones? This is especially surprising given that *Nanos* and *Pgc* levels will be higher in centrally located pole cells than in peripherally located ones as Slaidina and Lehmann (2017) showed that peripheral pole cells inherit less germ plasm than more central pole cells.

****Minor Concerns:****

1) Figure 1: There are some spelling mistakes in this figure.

2) Figure 2 (Panels B, D, and F): These panels don't indicate what protein or marker is being shown, unlike the other panels in other figures.

****Referee cross-commenting****

Reviewer 2: I agree with Reviewer 1 about the need for further controls for the RNAi experiments. I also think that more investigation into how *mira* over expression causes the observed phenotypes is desirable. However, I do not think that doing this is essential to support the present conclusions of the paper. Rather, it is something that would extend the study, and therefore I would not make it obligatory.

Reviewer 1: I do have a few serious concerns - in part about the RNAi experiments (phenotypes could be due to knockdown in the ovary or soma rather than in germ cells). I would also like to see some further investigation into how misexpression of *mira* causes the observed phenotypes.

Reviewer 2: I cannot see the other reviews so it is difficult to make specific comments. My

recollection however is that all three of us had relatively similar impressions of the manuscript, and that we all felt that only minor revisions were essential.

2. Significance:

Significance (Required)

Strength: The quality of the data is high - experiments are well executed and results are clear.

Limitation: The result that ectopic expression of single gene, *mira*, could determine whether pole cells remain physically separate from the somatic cells is surprising and there is no attempt to provide insight into why this might be the case and why *mira* in particular.

Advance: Provides reasonably compelling evidence that failure to repress transcription of somatic genes in pole cells is linked to failure to establish pole cells as distinct from soma.

Audience: The germ cell community

3. How much time do you estimate the authors will need to complete the suggested revisions:

Estimated time to Complete Revisions (Required)

(Decision Recommendation)

Between 1 and 3 months

No

Review #2

1. Evidence, reproducibility and clarity:

Evidence, reproducibility and clarity (Required)

Physical separation of germline and soma is usual in animal development and it is believed to be essential, but the mechanisms of how such separation is established and controlled remain unclear. Using genetic tools, Asaoka et al. produce *Drosophila* primordial germ cells that lack Pgc and overexpress Imp-alpha2, a key target of Nanos-mediated repression. Thus these cells are compromised for the two major pathways known to specify germline. By purifying these cells using FACS and comparative transcriptomics (both batch and single-cell) the authors establish that Pgc- Imp-alpha2 OE germ cells show increased levels of transcripts from a large number of genes whose expression is normally restricted to soma. These cells also exhibit changed morphology, in particular producing protuberances that enable them to intermingle with somatic cells, thus disturbing the usual separation between germline and soma. Ultimately the Pgc- Imp-alpha2 OE germ cells undergo cell death. Remarkably, the authors also show that these phenotypes can be largely rescued by knockdown of a single overexpressed somatic gene, namely *miranda* (*mira*).

- The paper's main strength is that it provides important new knowledge about how separation of germline and soma is achieved in *Drosophila* embryogenesis. It represents a major technical advance, in that it reports for the first time on global transcription changes in primordial germ cells that are compromised for major germ line-promoting pathways, and links those changes to morphological events. It also reports a major conceptual advance in that it implicates *Mira* as a regulator of germ line-soma segregation. Its main limitation is that, while it shows that ectopic *mira* drives the cellular morphology changes that enable the Pgc- Imp-alpha2 OE germ cells to intercalate with somatic cells, it does not provide a very specific model as to how *mira* might accomplish this. In neuroblast development *Mira* works in concert with three other proteins: *Pros*, *Stau*, and *Brat*. I perused Table S1, and found that *pros* is also upregulated in Pgc- Imp-alpha2 OE germ cells, actually to a greater extent than *mira*, and to a lesser degree *brat* is unregulated as well. At a minimum these results should be incorporated into the Discussion. OPTIONALLY, the conclusions of the paper could be extended by determining whether knockdown of *pros* also rescues the Pgc- Imp-alpha2 OE germ cell phenotypes.

- The Discussion should also address the fact that, while physical separation of germline and soma is widespread in evolution, *mira* does not have orthologues beyond insects. Thus other organisms must use different mechanisms, which may or may not be related to that described here, to accomplish this segregation.

- I believe that the conclusions of the authors are fully supported by the experimental data. The experiments are well documented, quantitative data are presented and statistical

analysis is appropriate. Sufficient methodological detail is provided to enable reproduction of the experiments.

- The manuscript could benefit from an additional minor round of copyediting as some grammatical and English usage errors are present.

****Referee cross-commenting****

Reviewer 2: I agree with Reviewer 1 about the need for further controls for the RNAi experiments. I also think that more investigation into how *mira* over expression causes the observed phenotypes is desirable. However, I do not think that doing this is essential to support the present conclusions of the paper. Rather, it is something that would extend the study, and therefore I would not make it obligatory.

Reviewer 1: I do have a few serious concerns - in part about the RNAi experiments (phenotypes could be due to knockdown in the ovary or soma rather than in germ cells). I would also like to see some further investigation into how misexpression of *mira* causes the observed phenotypes.

Reviewer 2: I cannot see the other reviews so it is difficult to make specific comments. My recollection however is that all three of us had relatively similar impressions of the manuscript, and that we all felt that only minor revisions were essential.

2. Significance:

Significance (Required)

The paper's main strength is that it provides important new knowledge about how separation of germline and soma is achieved in *Drosophila* embryogenesis. It represents a major technical advance, in that it reports for the first time on global transcription changes in primordial germ cells that are compromised for major germ line-promoting pathways, and links those changes to morphological events. It also reports a major conceptual advance in that it implicates *Mira* as a regulator of germ line-soma segregation. Its main limitation is that, while it shows that ectopic *mira* drives the cellular morphology changes that enable the *Pgc- Imp-alpha2* OE germ cells to intercalate with somatic cells, it does not provide a very specific model as to how *mira* might accomplish this.

The main audience for this paper would be basic researchers in developmental biology and cell biology.

3. How much time do you estimate the authors will need to complete the suggested revisions:

Estimated time to Complete Revisions (Required)

(Decision Recommendation)

Between 1 and 3 months

Yes

Review #3

1. Evidence, reproducibility and clarity:

Evidence, reproducibility and clarity (Required)

Repression of somatic gene expression in the early germ lineage is a critical mechanism to establish germ cell fate across species. Previously, work from this lab and others have demonstrated that Pgc and Nanos repress somatic gene expression in early *Drosophila* pole cells by repressing RNA polymerase II activity and inhibiting translation of the nuclear importin $\text{imp}\alpha 2$ respectively (Asaoka et al., 2019; Hanyu-Nakamura et al., 2008). In this study, Asaoka et al use RNA sequencing and microscopy to expand our understanding of the transcriptional and phenotypic consequences of somatic gene de-repression in pgc $\text{imp}\alpha 2$ OE mutants. The authors provide careful description of multiple pole cell phenotypes in pgc $\text{imp}\alpha 2$ OE mutants, including formation of cellular protrusions, mis-localization to the midgut primordium, and increased apoptosis. From the list of upregulated transcripts in pgc $\text{imp}\alpha 2$ OE mutants, the authors identify one transcript, mir, that is required for these defects. This work is carefully done and begins to describe a mechanism of how somatic gene expression impairs germ cell development. However, a few points need to be addressed, which can largely be rectified with changes to the text.

****Major Comments:****

The authors state this is the first time that repression of somatic transcription has been linked to the physical separation of germline and soma. Previous studies have reported pole cells "falling into" the yolk (Turner and Mahowald, 1976), which increases upon overexpression of germ cell-less, a factor associated with transcriptional repression in pole cells (Jongens et al., 1994; Leatherman et al., 2002). The authors should clarify how their phenotype is distinct from these previous reports. Furthermore, work in *C. elegans* embryos has shown that Nanos homologs repress oocyte and somatic gene expression by repressing the transcription factor Lin15B (Lee et al., 2017) and that loss of these factors leads to mis-localization of germ cells outside the somatic gonad (Subramaniam and Seydoux, 1999). The authors should acknowledge these studies.

Most graphs only show a single value, with no indication of variance. The methods state that "experiments for all figures and tables were repeated more than twice" so error bars and/or individual data points for these replicates should be shown.

Figures 2 and 3 demonstrate that pole cells mis-localize to the epithelium and have increased apoptosis in *pgc imp α 2* OE mutants relative to wild type. Migration defects and increased apoptosis are known phenotypes of *pgc* mutant pole cells (Deshpande et al., 2012; Hanyu-Nakamura et al., 2019; Nakamura et al., 1996). The data in this study should be compared to the previous findings. Furthermore, Figure 3 should demonstrate whether cleaved Dcp-1 levels in *pgc imp α 2* OE mutants are more severe than in *pgc* mutants alone. Previous work from these authors demonstrated that apoptosis was not more severe in *pgc imp α 2* OE mutants compared to *pgc* mutants (Asaoka et al., 2019). Any potential discrepancies between studies should be acknowledged.

****Minor Comments:****

In lines 80-82, the authors state "...Nanos and Polar granule component (Pgc), has been identified as factors required to repress somatic gene expression in pole cells." The earliest study demonstrating the role of Nanos for somatic gene repression should be cited as well (Deshpande et al., 1999).

In Figure 1b, there are several minor typos in the gene ontology terms such as "anaotmical", "formaiton involued", "Cell migaration"

In Table S1, a couple of gene symbols have been replaced by dates in the "Gene symbol" column (FBtr0088877 and FBtr0088876)

For Figure 2F, it is unclear what happens to the cell at the end of imaging because the marker dims dramatically. Does the cell stay in the somatic region over time? This data would be easier to interpret if the Moesin-GFP signal was brightened or another marker was shown.

In figures 4C and 5E, the authors demonstrate a correlation between frequency of pole cells expressing *mira* and formation of cellular protrusions and positions of the pole cells respectively. Can the authors determine if there is a correlation between *mira* levels and these phenotypes? A dose dependent effect could strengthen their argument that *mira* is responsible for these phenotypes.

The authors state in lines 225-226 "Here, we provide the first report of a molecular mechanism that controls spatial separation between pole cells and the soma". It would be more accurate that this mechanism "maintains" spatial separation. The mechanisms of pole cell cellularization, which initially separates pole cells from the soma, have been previously characterized.

In lines 256-257, the authors state "It is unlikely that the misexpression of widespread somatic genes in the pole cells affects their cell identity and results in cell death". This claim is not supported. The scRNA-seq data show that multiple somatic genes are mis-expressed in pole cells without *mira* expressions (clusters A and B), but the authors do not directly evaluate protrusions or apoptosis in these cells. Furthermore, there is no evidence demonstrating normal cell identity in these cells.

2. Significance:

Significance (Required)

Significance:

This work provides a conceptual advance to our understanding of the role of somatic gene repression in primordial germ cells and will be of interest to the field of germ cell biology. Previous work in *Drosophila* has established that *pgc* and *nos* are required to repress somatic gene expression in early pole cells (Deshpande et al., 1999; Martinho et al., 2004) and their mechanisms of repression (Asaoka et al., 2019; Hanyu-Nakamura et al., 2008). The phenotypes of *nos* and *pgc* mutants, such as migration defects and increased apoptosis, have been described and attributed in part to aberrant somatic gene expression (Asaoka-Taguchi et al., 1999; Deshpande et al., 2012; Hayashi et al., 2004; Kobayashi et al.,

1996; Nakamura et al., 1996). Although previous work demonstrated loss of *pgc* leads to aberrant miRNA expression and loss of Nanos, which are hypothesized to contribute to pole cell apoptosis (Deshpande et al., 2012; Hanyu-Nakamura et al., 2019), the somatic gene(s) that lead to these mutant phenotypes had not been definitively identified. This study provides a more global view of the gene misregulation that occurs when transcriptional quiescence is disrupted in pole cells. Additionally, the authors identify one misregulated gene, *mira*, that is required for increased cellular protrusion formation and apoptosis in these mutants, which begins to provide a mechanistic view of how somatic gene de-repression leads to specific phenotypic consequences. The mechanism identified in this work is analogous to work in *C. elegans*, which demonstrated that *nos* homologs prevents inappropriate transcriptional activation in primordial germ cells by inhibiting translation of the transcription factor Lin-15B (Lee et al., 2017), suggesting that the function of Nanos to repress aberrant transcription is conserved, although the exact mechanisms differ.

Reviewer areas of expertise: germ cell biology, RNA biology, transcriptional regulation

References

Asaoka M, Hanyu-Nakamura K, Nakamura A, Kobayashi S. 2019. Maternal Nanos inhibits Importin- α 2/Pendulin-dependent nuclear import to prevent somatic gene expression in the *Drosophila* germline. *PLoS Genet* 15:e1008090. doi:10.1371/journal.pgen.1008090

Asaoka-Taguchi M, Yamada M, Nakamura A, Hanyu K, Kobayashi S. 1999. Maternal Pumilio acts together with Nanos in germline development in *Drosophila* embryos. *Nat Cell Biol* 1:431-437. doi:10.1038/15666

Deshpande G, Calhoun G, Yanowitz JL, Schedl PD. 1999. Novel Functions of nanos in Downregulating Mitosis and Transcription during the Development of the *Drosophila* Germline. *Cell* 99:271-281. doi:10.1016/S0092-8674(00)81658-X

Deshpande G, Spady E, Goodhouse J, Schedl P. 2012. Maintaining sufficient Nanos is a critical function for polar granule component in the specification of primordial germ cells. *G3 (Bethesda)* 2:1397-1403. doi:10.1534/g3.112.004192

Hanyu-Nakamura K, Matsuda K, Cohen SM, Nakamura A. 2019. *Pgc* suppresses the zygotically acting RNA decay pathway to protect germ plasm RNAs in the *Drosophila* embryo. *Development* 146:dev167056. doi:10.1242/dev.167056

Hanyu-Nakamura K, Sonobe-Nojima H, Tanigawa A, Lasko P, Nakamura A. 2008. Drosophila Pgc protein inhibits P-TEFb recruitment to chromatin in primordial germ cells. Nature 451:730-733. doi:10.1038/nature06498

Hayashi Y, Hayashi M, Kobayashi S. 2004. Nanos suppresses somatic cell fate in Drosophila germ line. Proc Natl Acad Sci USA 101:10338-10342. doi:10.1073/pnas.0401647101

Jongens TA, Ackerman LD, Swedlow JR, Jan LY, Jan YN. 1994. Germ cell-less encodes a cell type-specific nuclear pore-associated protein and functions early in the germ-cell specification pathway of Drosophila. Genes Dev 8:2123-2136.

Kobayashi S, Yamada M, Asaoka M, Kitamura T. 1996. Essential role of the posterior morphogen nanos for germline development in Drosophila. Nature 380:708-711. doi:10.1038/380708a0

Leatherman JL, Levin L, Boero J, Jongens TA. 2002. germ cell-less acts to repress transcription during the establishment of the Drosophila Germ Cell Lineage. Current Biology 12:1681-1685. doi:10.1016/S0960-9822(02)01182-X

Lee C-YS, Lu T, Seydoux G. 2017. Nanos promotes epigenetic reprogramming of the germline by down-regulation of the THAP transcription factor LIN-15B. eLife 6:e30201. doi:10.7554/eLife.30201

Martinho RG, Kunwar PS, Casanova J, Lehmann R. 2004. A noncoding RNA is required for the repression of RNAPolIII-dependent transcription in primordial germ cells. Current Biology 14:159-165. doi:10.1016/j.cub.2003.12.036

Nakamura A, Amikura R, Mukai M, Kobayashi S, Lasko PF. 1996. Requirement for a noncoding RNA in Drosophila polar granules for germ cell establishment. Science 274:2075-2079. doi:10.1126/science.274.5295.2075

Subramaniam K, Seydoux G. 1999. nos-1 and nos-2, two genes related to Drosophila nanos, regulate primordial germ cell development and survival in Caenorhabditis elegans. Development 126:4861-4871. doi:10.1242/dev.126.21.4861

Turner FR, Mahowald AP. 1976. Scanning electron microscopy of *Drosophila* embryogenesis. *Dev Biol* 50:95-108. doi:10.1016/0012-1606(76)90070-1

3. How much time do you estimate the authors will need to complete the suggested revisions:

Estimated time to Complete Revisions (Required)

(Decision Recommendation)

Between 1 and 3 months

4. Review Commons values the work of reviewers and encourages them to get credit for their work. Select 'Yes' below to register your reviewing activity at Web of Science Reviewer Recognition Service (formerly Publons); note that the content of your review will not be visible on Web of Science.

Yes

Review #4

1. Evidence, reproducibility and clarity:

Evidence, reproducibility and clarity (Required)

In this manuscript, Asaoka et al., interrogate how segregation of the germline is achieved through the silencing of somatic genes. They observe that pole cells lacking Pgc and over-expressing *imp- α 2* (which is translationally repressed by Nanos in WT pole cells) exhibit an unusual morphology characterized by protrusions that penetrate the epithelial cells at the onset of gastrulation. By performing bulk RNA-sequencing, they compared *pgc- imp α 2OE* pole cells to WT pole cells and discovered that the *mira* gene, which is misexpressed in *pgc- imp α 2OE* pole cells, may be responsible for this abnormal morphology. They discover that *mira*-expressing pole cells in *pgc- imp α 2OE* mutants invade the midgut primordium and express the cell death marker *Dcp-1*. They further characterize the *pgc- imp α 2OE* pole cells using scRNA-sequencing and suggest that these *mira*-expressing pole cells contain less germ plasm components, and form on the periphery of the PGC rosette. Overall, this manuscript details an interesting question, and is well written and easy to follow. However, we find that the claims in the paper are significantly overstated, as there is no direct

evidence that the germ cell death described here is due to the loss of soma/germline segregation. *pgc- imp α 2OE* pole cells expressing *mira* do not phenocopy *Nanos* mutant germ cells, and have not been shown to transform pole cells into somatic cells as is described for *Nanos* (Hayashi et al., PMID 15240884). Instead, it is possible that *mira*-expressing pole cells simply migrate early, and it is this precocious migration that results in the observed phenotypes. While this early migration does disrupt the physical separation between the germline and soma as the authors highlight, it is not clear if the physical intermingling of germline/soma is the cause of the cell death, or just a consequence of the early migration. Additional experiments are necessary to fully elucidate if expression of *mira* is sufficient to induce cell death, or if additional ectopic somatic gene expression contributes to the observed phenotypes.

****Major points:****

1. The conclusions drawn throughout the manuscript tend to be overstated for the strength of the findings. The authors state that they describe a critical role for the *Nanos/Pgc*-dependent somatic gene expression in the spatial segregation of the germline from the soma. However *pgc- imp α 2OE* pole cells do not phenocopy *nos/pgc* mutant pole cells (Asaoka et al., 2019). *imp α 2OE* in pole cells has no effect on mitosis, apoptosis, or migration, and *pgc- imp α 2OE* pole cells do not transform germ cells as has been described for *Nanos*. Thus, while *Nanos* and *Pgc* may act together as a 'double-lock' mechanism to ensure inhibition of somatic gene expression, *pgc- imp α 2OE* pole cells do not fully disrupt this double lock. From the data, it is not clear what role *mira* plays in this process, as a causal relationship between *mira* expression, germline/soma intermingling, and germ cell death has not been established here.
2. Is ectopic expression of *mira* in WT pole cells sufficient to induce pole cell protrusions and *Dcp-1* expression? And if so, are these phenotypes more likely to occur in peripheral vs central pole cells? Can the authors ectopically express a *mira* construct with a *nanos* 3'UTR (for expression in the germline) in both the presence and absence of the *imp α 2-nos* 3'UTR construct made in the Asaoka et al., 2019 Plos Genetics paper? This is particularly useful as *imp α 2* expression alone has no significant effects until gametogenesis. If ectopic expression of *mira* in an otherwise WT germline induces both the protrusions, midgut primordium invasion, and the cell death phenotypes, this would show sufficiency.
3. Error bars on graphs in figures 4 and 5 would be helpful in depicting biological differences between embryos. Rather than plotting the % of pole cells expressing each mRNA, can you instead plot the % of pole cells per embryo that expresses each mRNA,

with error bars indicating the biological variation between embryos? Additional information on how many embryos were counted in each experiment would also be helpful.

4. The authors state that *pgc- imp α 2OE* pole cells that express *mira* will eventually die. Can the authors count the number of pole cells that coalesce with the gonad in *pgc- imp α 2OE* mutants vs WT embryos? If there are less pole cells that coalesce with the gonad in *pgc- imp α 2OE* embryos, does knockdown of *mira* rescue this loss?

5. There are very few cells in the scRNA-sequencing analysis of the *pgc- imp α 2OE* pole cells. While SMART-seq can yield libraries of higher complexity/depth than 10x, the low cell count in this experiment makes it difficult to cluster cells with accuracy. More *in vivo* validation is necessary to make conclusions based on this dataset:

a. Can the authors validate the scRNA-sequencing result by performing *in situ* for *mira* RNA and a germline RNA (such as *gcl* or *osk* - as these show the strongest phenotype in the sequencing) then quantify the amount of each RNA in pole cells? If so, there should be an anti-correlation between *mira* expression and *osk* expression. Does this also correspond to whether the pole cell is peripheral or central?

b. How was the data in Fig. 5E measured and calculated? Were *in situ* performed for *mira*? If so, representative images should be shown and a description of how expression was calculated and what probes were used should be added to the methods.

c. Representative images for figure 5F should also be shown.

****Minor points:****

1. The authors describe GO term analysis of WT vs *pgc- imp α 2OE* pole cells in which downregulated transcripts in *pgc- imp α 2OE* pole cells are enriched for terms unrelated to germline development, suggesting that germline components are normally retained in these mutants. However accurate interpretation of GO term enrichment is highly dependent on using the correct background gene set. What control gene set did the authors use to perform GO term analysis (this information was not in the materials and methods)? Was the WT pole cell expression data used as background? This might be a good dataset to use to set the background, if not used previously. If a background gene set was not previously specified, it is essential to perform the analysis with an appropriate background gene set.

2. It would be helpful to keep the formatting of the graphs the same across experiments. Figures 4A, 5E, and 5F have the significance denoted with a horizontal bar and an asterisks. A similar horizontal bar might also be helpful in figures 4C, D, F, and H, rather than the

diagonal bar.

3. A similar pole cell protrusion phenotype has been observed previously in wildtype and more extensively and precociously in Jafrac1 (Prx2) and DE-Cadherin mutant embryos (Kunwar et al., 2008 PMID: 18824569 and DeGennaro et al., 2011 PMID 21316590) as part of the normal transition of pole cells toward migratory behavior. Is Jafrac1 or DE-Cadherin downregulated in *pgc- imp α 2OE* mutant embryos/pole cells? Do increased DE-Cadherin levels rescue the protrusion phenotype?

2. Significance:

Significance (Required)

Overall, this manuscript details an interesting question, and is well written and easy to follow. However, we find that the claims in the paper are significantly overstated, as there is no direct evidence that the germ cell death described here is due to the loss of soma/germline segregation

3. How much time do you estimate the authors will need to complete the suggested revisions:

Estimated time to Complete Revisions (Required)

(Decision Recommendation)

Cannot tell / Not applicable

No

Full Revision

Manuscript number: RC-2025-03075

Corresponding author(s): Miho Asaoka, Satoru Kobayashi

1. General Statements [optional]

We would like to thank the reviewers for their valuable and insightful comments. We have carefully considered all the suggestions and revised the manuscript accordingly, with some new experimental results. The goal of this work is to identify molecular mechanisms controlling physical separation between the germline and soma, which is an evolutionarily conserved but largely unexplored phenomenon. Our data demonstrate that this separation is maintained by repressing a somatic gene, *mira*, via a Nanos/Pgc-dependent double repression mechanism and suggest its possible role in preventing germ cell death. We believe that the revised manuscript now communicates these findings more effectively.

1) Reviewer 1 (major Concern 1) stated that “*Figure 2: The authors analyze pole cell protrusions during early embryogenesis, but it remains unclear how these morphological changes in the pole cells impact the end result of gonad formation or germ cell development. Further discussion or experimentation linking these early events to later defects in germline maturation would strengthen the developmental relevance of these findings.*”

Our response

We thank the reviewer for this valuable comment. We could not analyze the phenotypes specific to *pgc impa2OE* pole cells beyond stage 9, because, from stage 10 onwards, apoptosis and ectopic mitosis are induced in pole cells by *pgc* mutation alone [1, 2]. Our previous work demonstrated that *pgc impa2OE* embryos develop into agametic adults at a higher frequency than either *pgc*⁻ or *impa2OE* embryos [1], suggesting that pole cells that survive beyond gastrulation in *pgc impa2OE* embryos show abnormalities through *pgc*-independent mechanisms. We have added the sentences to the *Results* (P8 L181–L185) and the *Discussion* sections detailing the same (P13 L316–P14 L325).

2) Reviewer 1 (major concern 2) stated that “*Figure 3: The authors suggest that the pole cells forming protrusions are the same ones that later invade the midgut epithelium and undergo apoptosis. However, there is no direct experimental evidence connecting these two observations. While live imaging using Moesin-GFP is limited by the loss of the protrusion phenotype as PGCs migrate, staining for Hid or Atg1, autophagy-related proteins known to induce cleaved Dcp-1, could provide a clearer link. This is especially important given the transcriptomic upregulation of autophagy genes in early pole cells, potentially allowing identification of protruding cells destined for apoptosis.*”

Our response

We appreciate your comment. In this study, we focused on demonstrating that *mira* misexpression is required for these phenotypes including apoptosis. This is the first report uncovering the mechanisms that maintain the spatial segregation between germline and soma. As mentioned by the reviewer 2, identification of the downstream gene for *mira* function must be beyond the scope of this manuscript. Thus, we decided that this experiment is reserved for future investigation.

3) Reviewer 1 (major concern 3) stated that “*The authors utilize the MatGal4 driver, which drives expression broadly during oogenesis and the embryo (including the soma). This raises the concern that pole cell phenotypes observed with mira RNAi could reflect effects from depletion of mira in the oocyte or in the somatic cells at the posterior, which could influence pole cell behavior (ZGA in soma precedes ZGA in pole cells). How do the authors distinguish between PGC-specific effects and possible somatic contributions? This issue may also explain why central pole cells appear unaffected, while peripheral pole cells, those in closest proximity to the soma, exhibit defects. Overexpressing mira could help to determine whether the phenotype is intrinsic to pole cells or influenced by their somatic microenvironment. If overexpression fails to induce protrusions in central pole cells, this would suggest a non-cell-autonomous role of the soma in driving the observed behavior.*”

Our response

Thank you for your comment. We agree with your points. To distinguish between PGC-specific effects and possible somatic contributions, we performed pole cell transplantation experiments. We transplanted *pgc impα2OE* or *w* (control) pole cells into normal host embryos (*y w*) at the blastoderm stage and followed their fate until gastrulation (stages 7–9). Our results indicate that the double-lock mechanism achieved by Pgc and the depletion of *Impα2* is autonomously required in pole cells to prevent their intermingling with epithelial cells in the midgut primordium. We have added these data in Table 1 and the corresponding statement in the *Result* section (P7 L161–164). We have also now described the transplantation method in the *Materials and methods* section (P19 L442–456).

4) Reviewer 1 (major concern 4) stated that “*While the authors show that mira (either in pole cells or soma, see above) is necessary for the protrusion phenotype in peripheral pole cells, the mechanism by which mira promotes these morphological changes is unclear. Functional studies examining how mira contributes to membrane dynamics, polarity, or cell motility could clarify its role. Additionally, the study focuses solely on expression of mira RNA, without addressing mira protein localization or function. What does Mira protein do (in other contexts) and how might this function relate to a function in pole cells?*”

Our response

Thank you for your comment. We completely agree that investigating how *mira* contributes to membrane dynamics, polarity, or motility as well as examining the localization and function of Mira protein, would be important directions for future research. According to the statement

of Reviewer 2 in the cross-comments, such analyses are not essential to support our current conclusions. We therefore prefer not to pursue these experiments within the framework of this revision.

5) Reviewer 1 (major concern 5) stated that “*What happens to the central pole cells in *pgc-impα2OE* embryos? Do they survive and populate the gonad? This should be addressed.*”

Our response:

We thank the reviewer for raising this important question. In the original manuscript, we showed that in *pgc-impα2OE* embryos, peripheral pole cells misexpressed *mira* and formed protrusions, whereas central pole cells maintained *mira* repression and retained their normal spherical morphology. These results suggest that pole cells surviving beyond stage 10 are primarily derived from the central pole cells. Thus, we speculate that the central pole cells showed the later phenotypes observed in *pgc⁻ impα2OE* pole cells after stage 10 [1]. We have now incorporated this information in the *Discussion* section (P13 L316–P14 L325).

6) Reviewer 1 (major concern 6) stated that “*Why would Nanos and Pgc be sufficient to repress *mira* in the peripheral pole cells but not in the central ones? This is especially surprising given that Nanos and Pgc levels will be higher in centrally located pole cells than in peripherally located ones as Slaidina and Lehmann (2017) showed that peripheral pole cells inherit less germ plasm than more central pole cells.*”

Our response

Thank you for pointing this out. Our original wording was misleading and did not accurately reflect our intended interpretation. Nanos and Pgc are critical for *mira* repression in the peripheral pole cells, which inherit lower amounts of germ plasm, whereas in the central pole cells, *mira* is repressed mainly by an additional germ-plasm component(s). Please see the revised sentences in the *Discussion* section for better clarity (P13 L306–L310).

7) Reviewer 1, Reviewer 2, and Reviewer 3 stated as follows: “*Figure 1: There are some spelling mistakes in this figure*” (minor concern 1 by Reviewer 1). “*The manuscript could benefit from an additional minor round of copyediting as some grammatical and English usage errors are present*” (comment by Reviewer 2). “*In Figure 1b, there are several minor typos in the gene ontology terms such as ‘anaotmical’, ‘formaiton involued’, ‘Cell migaration’*” (minor concern 2 by Reviewer 3).

Our response

Thank you for pointing this out. We have now carefully checked the figure to ensure they are error-free.

8) Reviewer 1 (minor concern 2) stated that “*Figure 2 (Panels B, D, and F): These panels don’t indicate what protein or marker is being shown, unlike the other panels in other figures.*”

Our response

Thank you for highlighting this. We have revised **Fig 2** by adding the marker names below or side of the **panels B, D, and F**.

9) Reviewer 2 stated that “*Its main limitation is that, while it shows that ectopic mira drives the cellular morphology changes that enable the Pgc- Imp-alpha2 OE germ cells to intercalate with somatic cells, it does not provide a very specific model as to how mira might accomplish this. In neuroblast development Mira works in concert with three other proteins: Pros, Stau, and Brat. I perused Table S1, and found that pros is also upregulated in Pgc- Imp-alpha2 OE germ cells, actually to a greater extent than mira, and to a lesser degree brat is unregulated as well. At a minimum these results should be incorporated into the Discussion. OPTIONALLY, the conclusions of the paper could be extended by determining whether knockdown of pros also rescues the Pgc- Imp-alpha2 OE germ cell phenotypes.*”

Our response

We thank the reviewer for this helpful comment. We have added the following sentence to the *Discussion* section (**P12 L281–287**): “Mira acts together with three other proteins, Prospero (Pros), Staufen (Stau), and Brain tumor (Brat), in neuroblast [3]. Interestingly, in *pgc⁻ impa2OE* pole cells, the expression of *pros* and *brat* mRNAs was upregulated compared to that in normal pole cells (Table S1), and *stau* mRNA was also detected at a significant level [Total *stau* expression (sum of three annotated isoforms): 31.6 ± 14.7 (TPM \pm SEM, three biological replicates)]. It is necessary to investigate whether these three proteins act together with Mira for protrusion formation of *pgc⁻ impa2OE* pole cells.”

Regarding the optional suggestion to examine whether *pros* knockdown rescues the *pgc- impa2OE* phenotype, we agree that such experiments would be valuable. However, these would considerably extend the scope of the present study and are not essential to support our current conclusions. We therefore prefer to leave this point as an interesting direction for future investigation.

10) Reviewer 2 stated that “*The Discussion should also address the fact that, while physical separation of germline and soma is widespread in evolution, mira does not have orthologues beyond insects. Thus other organisms must use different mechanisms, which may or may not be related to that described here, to accomplish this segregation.*”

Our response

Thank you for your comment. We have now added the following sentence to the *Discussion* section (**P14 L331–335**): “Although no orthologs of *mira* have been reported outside insects (FlyBase, <https://flybase.org/>), other organisms may achieve a similar outcome by repressing different somatic genes through Nanos and global transcriptional silencing mechanisms. Further studies will provide new insights into both the conserved and divergent mechanisms underlying this process.”

11) Reviewer 3 (major concern 1) stated that “*The authors state this is the first time that repression of somatic transcription has been linked to the physical separation of germline and soma. Previous studies have reported pole cells ‘falling into’ the yolk (Turner and Mahowald, 1976), which increases upon overexpression of germ cell-less, a factor associated with transcriptional repression in pole cells (Jongens et al., 1994; Leatherman et al., 2002). The authors should clarify how their phenotype is distinct from these previous reports.*”

Our response

We appreciate this comment by the reviewer. The invasion phenotype of *pgc⁻impa2OE* pole cells is distinct from the previously described “falling into the yolk” phenotype. We examined the percentage of pole cells migrating through the midgut wall into the hemocoel (yolk region) and found that it was low in *pgc⁻impa2OE* pole cells and did not increase compared to that in *y w* controls. We have added these data in S1C Fig and clarified this point in the *Results* section (P6 L144–150). We have also added legend for S1C Fig on P25 L571–L577.

12) Reviewer 3 (major concern 2) stated that “*Furthermore, work in C. elegans embryos has shown that Nanos homologs repress oocyte and somatic gene expression by repressing the transcription factor Lin15B (Lee et al., 2017) and that loss of these factors leads to mis-localization of germ cells outside the somatic gonad (Subramaniam and Seydoux, 1999). The authors should acknowledge these studies.*”

Our response

We thank the reviewer for this suggestion. We have now cited these studies in the *Introduction* (Lee et al., 2017; P4 L81) and *Discussion* sections (Subramaniam and Seydoux, 1999; P14 L329) to acknowledge their relevance to our work.

13) Reviewer 3 and Reviewer 4 stated as follows: “*Most graphs only show a single value, with no indication of variance. The methods state that ‘experiments for all figures and tables were repeated more than twice’ so error bars and/or individual data points for these replicates should be shown*” (**major concern 3 by Reviewer 3**). “*Error bars on graphs in figures 4 and 5 would be helpful in depicting biological differences between embryos. Rather than plotting the % of pole cells expressing each mRNA, can you instead plot the % of pole cells per embryo that expresses each mRNA, with error bars indicating the biological variation between embryos? Additional information on how many embryos were counted in each experiment would also be helpful*” (**major point 3 by Reviewer 4**).

Our response

We thank the reviewers for these helpful suggestions. Following reviewer 4’s recommendation, we have revised all bar graphs in the original manuscript into beeswarm plots combined with boxplots in the revised version. We have also added the number of embryos analyzed in each experiment, indicated as (N) below the graphs. The updated figures include Figs 2C, 2E, 3C, 3D, 4A, 4C, 4D, 4F, 4H, 5E, and 5F. In addition, we have

revised the corresponding **figure legends** (P35 L815–P39 L914) to clarify how the data are presented.

14) Reviewer 3 (major concern 4) stated that “*Figures 2 and 3 demonstrate that pole cells mis-localize to the epithelium and have increased apoptosis in *pgc impα2* OE mutants relative to wild type. Migration defects and increased apoptosis are known phenotypes of *pgc* mutant pole cells (Deshpande et al., 2012; Hanyu-Nakamura et al., 2019; Nakamura et al., 1996). The data in this study should be compared to the previous findings. Furthermore, Figure 3 should demonstrate whether cleaved *Dcp-1* levels in *pgc impα2* OE mutants are more severe than in *pgc* mutants alone. Previous work from these authors demonstrated that apoptosis was not more severe in *pgc impα2* OE mutants compared to *pgc* mutants (Asaoka et al., 2019). Any potential discrepancies between studies should be acknowledged.*”

Our response

We thank the reviewer for raising this important point. To clarify the relationship between our current findings and previous works, we have revised the manuscript at two points.

(i) We have added the following sentences to the *Results* section (P8 L180–L185): “Thus, *pgc⁻ impα2OE* pole cells undergo cell death during gastrulation (stages 7–9). Notably, *pgc⁻* pole cells show apoptotic phenotype only after stage 10 [2] and that apoptosis is never detected in *impα2OE* pole cells [1], suggesting that the early-stage cell death of *pgc⁻ impα2OE* pole cells is driven by mechanisms distinct from those underlying late-stage apoptosis in *pgc⁻* pole cells.”

(ii) We have added the following sentences to the *Discussion* section (P13 L319–P14 L325): “These abnormalities are distinct from the ones caused by the absence of *pgc*. It has been reported that *pgc* mutation causes *nanos* degradation in pole cells and consequently induces abnormalities in mitosis, apoptosis and migration of pole cells after stage 10 onward [2]. Combining these early (stages 5–9) and late (stage 10 onward) abnormalities results in lower fertility of *pgc⁻ impα2OE* embryos than that of *pgc⁻* mutants or *impα2OE* embryos [1]”

These revisions clearly state the relationship between current findings (stages 6–9) and previously reported phenotypes (after stage 10).

15) Reviewer 3 (minor concern 1) stated that “*In lines 80-82, the authors state “...Nanos and Polar granule component (Pgc), has been identified as factors required to repress somatic gene expression in pole cells.” The earliest study demonstrating the role of Nanos for somatic gene repression should be cited as well (Deshpande et al., 1999).*”

Our response

As per the reviewer’s suggestion, we have cited the study reported by *Deshpande et al.* (1999) in the *Introduction* section (P4 L85 in the revised version).

16) Reviewer 3 (minor concern 3) stated that “*In Table S1, a couple of gene symbols have been replaced by dates in the “Gene symbol” column (FBtr0088877 and FBtr0088876).*”

Our response

We thank the reviewer for pointing this out. We have corrected the “Gene symbol” for FBtr0088877 and FBtr0088876 in S1 Table.

17) Reviewer 3 (minor concern 4) stated that *“For Figure 2F, it is unclear what happens to the cell at the end of imaging because the marker dims dramatically. Does the cell stay in the somatic region over time? This data would be easier to interpret if the Moesin-GFP signal was brightened or another marker was shown.”*

Our response

Following the reviewer’s recommendation, we have now added a brightened and enlarged image of the pole cells at time=2’40” in the lower right corner in Fig 2F. In this image, it is clear that *pgc impα2 OE* pole cells remained within the epithelial layer of the midgut primordium, but their *Moesin-GFP* signal remarkably decreased compared to that in pole cells located on the surface of the embryo. We have also now stated this in the legend of Fig 2 (P36 L829–L833).

18) Reviewer 3 (minor concern 5) stated that *“In figures 4C and 5E, the authors demonstrate a correlation between frequency of pole cells expressing mira and formation of cellular protrusions and positions of the pole cells respectively. Can the authors determine if there is a correlation between mira levels and these phenotypes? A dose dependent effect could strengthen their argument that mira is responsible for these phenotypes.”*

Our response

We agree that examining a dose-dependent effect of *mira* on protrusion formation and pole cell positioning could strengthen our argument. However, such experiments are technically challenging. In the analyses for Figs 4C and 5E, we simultaneously assessed *mira* mRNA expression by *in situ* hybridization and pole cell morphology by anti-Vasa staining. In this method, we minimized proteinase K treatment during *in situ* hybridization to preserve Vasa protein and pole cell morphology, at the expense of probe accessibility. Under this experimental condition, we cannot quantify *mira* expression levels.

19) Reviewer 3 (minor concern 6) stated that *“The authors state in lines 225-226 “Here, we provide the first report of a molecular mechanism that controls spatial separation between pole cells and the soma”. It would be more accurate that this mechanism “maintains” spatial separation. The mechanisms of pole cell cellularization, which initially separates pole cells from the soma, have been previously characterized.”*

Our response

Following the reviewer’s recommendation, we have revised this sentence as follows (P11 L251–L253 in the revised version).

“Here, we provide the first report of a molecular mechanism that controls the spatial separation between pole cells and the soma (Fig 6).”

“Here, we provide the first report of a molecular mechanism that **maintains** the spatial separation between pole cells and the soma (Fig 6).”

20) Reviewer 3 (minor concern 7) state that “*In lines 256-257, the authors state "It is unlikely that the misexpression of widespread somatic genes in the pole cells affects their cell identity and results in cell death". This claim is not supported. The scRNA-seq data show that multiple somatic genes are mis-expressed in pole cells without *mira* expressions (clusters A and B), but the authors do not directly evaluate protrusions or apoptosis in these cells. Furthermore, there is no evidence demonstrating normal cell identity in these cells.*”

Our response

We agree with the reviewer’s comments. Therefore, we have **deleted the sentence** from the third paragraph of the *Discussion* section (**P12 L289–290 in the revised version**), **Fig S2, and its figure legend**.

21) Reviewer 4 (major point 1) stated that “*The conclusions drawn throughout the manuscript tend to be overstated for the strength of the findings. The authors state that they describe a critical role for the Nanos/Pgc-dependent somatic gene expression in the spatial segregation of the germline from the soma. However *pgc- impα2OE* pole cells do not phenocopy *nos/pgc* mutant pole cells (Asaoka et al., 2019). *impα2OE* in pole cells has no effect on mitosis, apoptosis, or migration, and *pgc- impα2OE* pole cells do not transform germ cells as has been described for Nanos. Thus, while Nanos and Pgc may act together as a 'double-lock' mechanism to ensure inhibition of somatic gene expression, *pgc- impα2OE* pole cells do not fully disrupt this double lock. From the data, it is not clear what role *mira* plays in this process, as a causal relationship between *mira* expression, germline/soma intermingling, and germ cell death has not been established here.*”

Our response

The differences from *nanos/pgc* mutant phenotypes:

These can be explained by two factors. First, Nanos regulates additional targets beside *impα2* mRNA. For example, repressing *cyclin B* and *hid* translation suppresses mitosis and apoptosis in pole cells, respectively [4, 5], while migratory defects in late embryogenesis arise secondary in an apoptosis-dependent manner [6]. Thus, *impα2* overexpression alone cannot recapitulate these Nanos-specific functions, even when the double-lock mechanism is fully disrupted.

Second, the timing of phenotypes is critical. The phenotypes we describe here occur at stages 6–9, earlier than those reported in *pgc*⁻ mutants (stage 10 onward). In *pgc*⁻ mutants, the reduction of Nanos protein level from stage 10 accounts for mitotic activation, apoptosis, and migration failure [2]. Consistently, *pgc*⁻ *impα2OE* pole cells also exhibit ectopic mitosis, apoptosis, and migration defects at stages 10–16, with comparable frequencies to *pgc*⁻ mutants [1].

In summary, phenotypic differences from *nanos*⁻ and *pgc*⁻ mutants reflect (i) Nanos-specific regulatory targets beyond *impα2*, and (ii) differences in the developmental stages when these phenotypes appear. We believe this reconciles the reviewer's concern.

Regarding the reviewer's concern on *mira*:

To avoid overstatement, we have revised the last sentence of the first paragraph of the *Discussion* (P11 L258–L260) as follows:

“In the absence of these two repression mechanisms, pole cells **misexpressed** *mira* and **intermingled** with the soma eventually die.”

↓

“In the absence of these two repression mechanisms, pole cells **misexpress** *mira*, **intermingle** with the soma, **and** eventually die.”

This change removes the implication of a direct causal chain among these events.

22) Reviewer 4 (major point 2) stated that “*Is ectopic expression of mira in WT pole cells sufficient to induce pole cell protrusions and Dcp-1 expression? And if so, are these phenotypes more likely to occur in peripheral vs central pole cells? Can the authors ectopically express a mira construct with a nanos 3'UTR (for expression in the germline) in both the presence and absence of the impα2-nos3'UTR construct made in the Asaoka et al., 2019 Plos Genetics paper? This is particularly useful as impα2 expression alone has no significant effects until gametogenesis. If ectopic expression of mira in an otherwise WT germline induces both the protrusions, midgut primordium invasion, and the cell death phenotypes, this would show sufficiency.*”

Our response

We agree with the reviewer's questions. However, addressing whether *mira* alone is sufficient to drive the protrusion, intermingling, and cell death phenotypes will be an important future direction. We therefore prefer not to pursue these experiments within the framework of this revision.

23) Reviewer 4 (major point 4) stated that “*The authors state that pgc- impα2OE pole cells that express mira will eventually die. Can the authors count the number of pole cells that coalesce with the gonad in pgc- impα2OE mutants vs WT embryos? If there are less pole cells that coalesce with the gonad in pgc- impα2OE embryos, does knockdown of mira rescue this loss?*”.

Our response

In our previous study [1], we showed that the percentage of pole cells that coalesce with the embryonic gonads was significantly lower in *pgc*⁻ *impα2OE* embryos than in wild-type embryos. However, this reduction was also observed in *pgc*⁻ embryos at a comparable frequency. These findings indicate that the reduction reflects the *pgc* mutation (in which Nanos levels decrease from stage 10 onward) rather than *mira*-dependent cell death.

To avoid overstatement, we have revised the last sentence of the first paragraph of the *Discussion* (P11 L258–L260) as follows. This change removes the implication of a direct causal link between *mira* misexpression and cell death in *pgc⁻ impα2OE* pole cells.

“In the absence of these two repression mechanisms, pole cells **misexpressed** *mira* and **intermingled** with the soma eventually die.”

↓

“In the absence of these two repression mechanisms, pole cells **misexpress** *mira*, **intermingle** with the soma, **and** eventually die.”

24) Reviewer 4 (major point 5) stated that “*There are very few cells in the scRNA-sequencing analysis of the pgc- impα2OE pole cells. While SMART-seq can yield libraries of higher complexity/depth than 10x, the low cell count in this experiment makes it difficult to cluster cells with accuracy. More in vivo validation is necessary to make conclusions based on this dataset:*
a. *Can the authors validate the scRNA-sequencing result by performing in situs for mira RNA and a germplasm RNA (such as gcl or osk- as these show the strongest phenotype in the sequencing) then quantify the amount of each RNA in pole cells? If so, there should be an anti-correlation between mira expression and osk expression. Does this also correspond to whether the pole cell is peripheral or central?*”

Our response

We thank the reviewer for this question. The correlation (anti-correlation) can be validated by *in situ* hybridization with probes for *mira* mRNA and germ-plasm component RNAs simultaneously. We have not performed simultaneous *in situ* hybridization. However, in our scRNA-seq analyses, we found that the expression levels of five germ-plasm component mRNAs (*gcl*, *nanos*, *osk*, *piwi*, and *Prx2*) negatively correlate with *mira* expression (Pearson correlation coefficient = -0.66, -0.46, -0.57, -0.58, -0.70, respectively; $P < 0.01$, two-sided Student’s *t*-test). These results strongly support our conclusion that *mira* is ectopically expressed in the *pgc⁻ impα2OE* pole cells that inherit fewer germ-plasm components than the other pole cells. We have added these results showing the correlations between *mira* and germ-plasm component mRNAs to the *Results* section (P10 L233–237) in the revised manuscript.

b. *How was the data in Fig. 5E measured and calculated? Were in situs performed for mira? If so, representative images should be shown and a description of how expression was calculated and what probes were used should be added to the methods.*

Our response

Yes, we performed triple-staining for *mira* mRNA, Vasa protein, and nuclei. Detailed staining methods and probes for *mira* are described in the *in situ* hybridization section of *Materials and methods* (P17 L390–P18 L424 and P16 L384–P17 L388, respectively). In addition, we have now added a description of how *mira* expression was quantified (P18 L424–431). We

have also included a confocal image showing that *mira* mRNA is enriched in peripheral pole cells in **S2A Fig** and described this in the corresponding figure legend (**P25 L579–P26 L586**).

c. Representative images for figure 5F should also be shown.

Our response

Thank you for your comment. We have added a confocal image showing that peripheral pole cells form protrusions (whereas central pole cells do not) in **S2B Fig** and described this in the corresponding figure legend (**P25 L583–P26 L587**).

25) Reviewer 4 (minor point 1) stated that “*The authors describe GO term analysis of WT vs *pgc- impα.2OE* pole cells in which downregulated transcripts in *pgc- impα.2OE* pole cells are enriched for terms unrelated to germline development, suggesting that germline components are normally retained in these mutants. However accurate interpretation of GO term enrichment is highly dependent on using the correct background gene set. What control gene set did the authors use to perform GO term analysis (this information was not in the materials and methods)? Was the WT pole cell expression data used as background? This might be a good dataset to use to set the background, if not used previously. If a background gene set was not previously specified, it is essential to perform the analysis with an appropriate background gene set.*”

Our response

We thank the reviewer for pointing out this important issue. Following the reviewer’s suggestions, we re-performed the GO enrichment analysis using the filtered transcript set that is used in our RNA-seq analyses [16,009 transcripts (8305 gene)] as the background gene set. Accordingly, we have updated the results in **Figs. 1B, 1D** and **Tables S1–S3**.

In the latest version of Metascape (Database & MSBio release 2025-07-01, updated on July 3, 2025), the GO term “Generation of neurons”, which we previously highlighted, no longer appeared as a top-level representative category. This is due to changes in clustering and representative term selection. In the revised manuscript, we therefore focused on the related GO terms “Neuroblast differentiation” and “Central nervous system development” and selected four candidate mRNAs (*ac*, *mira*, *sc*, and *tll*). After validation by *in situ* hybridization, we focused on *mira*. These revisions have been incorporated into the *Results* section (**P8 L196–P9 L205**), **Fig 4A**, and the corresponding figure legend (**P37 L864–865**). We have also added the description about probes for *ac* in the *Materials and methods* section (**P16 L378–L381**).

26) Reviewer 4 (minor point 2) stated that “*It would be helpful to keep the formatting of the graphs the same across experiments. Figures 4A, 5E, and 5F have the significance denoted with a horizontal bar and an asterisks. A similar horizontal bar might also be helpful in figures 4C, D, F, and H, rather than the diagonal bar.*”

Our response

Following the reviewer's recommendation, we have revised **Figs 4C, 4D, 4F, and 4H** with a horizontal bar and asterisks.

27) Reviewer 4 (minor point 3) stated that “A similar pole cell protrusion phenotype has been observed previously in wildtype and more extensively and precociously in *Jafrac1* (*Prx2*) and *DE-Cadherin* mutant embryos (Kunwar et al., 2008 PMID: 18824569 and DeGennaro et al., 2011 PMID 21316590) as part of the normal transition of pole cells toward migratory behavior. Is *Jafrac1* or *DE-Cadherin* downregulated in *pgc- imp α 2OE* mutant embryos/pole cells? Do increased *DE-Cadherin* levels rescue the protrusion phenotype?”

Our response

In our RNA-seq analyses, we did not detect differences in *Prx2/jafrac1* and *shg* (encoding *DE-Cadherin*) mRNA levels between *pgc- imp α 2OE* and control pole cells. Thus, we did not examine, although we cannot exclude a possibility that their protein levels were altered. We consider it an important future direction to test the involvement of *Prx2/Jafrac1* and *DE-Cadherin* proteins.

References

1. Asaoka M, Hanyu-Nakamura K, Nakamura A, Kobayashi S. Maternal Nanos inhibits Importin- α 2/Pendulin-dependent nuclear import to prevent somatic gene expression in the *Drosophila* germline. *PLoS Genet.* 2019;15(5):e1008090. Epub 2019/05/16. doi: 10.1371/journal.pgen.1008090. PubMed PMID: 31091233; PubMed Central PMCID: PMC6519790.
2. Hanyu-Nakamura K, Matsuda K, Cohen SM, Nakamura A. Pgc suppresses the zygotically acting RNA decay pathway to protect germ plasm RNAs in the *Drosophila* embryo. *Development (Cambridge, England).* 2019;146(7). Epub 20190404. doi: 10.1242/dev.167056. PubMed PMID: 30890569.
3. Knoblich JA. Asymmetric cell division: recent developments and their implications for tumour biology. *Nat Rev Mol Cell Biol.* 2010;11(12):849–60. doi: 10.1038/nrm3010. PubMed PMID: 21102610; PubMed Central PMCID: PMC3941022.
4. Asaoka-Taguchi M, Yamada M, Nakamura A, Hanyu K, Kobayashi S. Maternal Pumilio acts together with Nanos in germline development in *Drosophila* embryos. *Nat Cell Biol.* 1999;1(7):431–7. PubMed PMID: 10559987.
5. Sato K, Hayashi Y, Ninomiya Y, Shigenobu S, Arita K, Mukai M, et al. Maternal Nanos represses *hid/skl*-dependent apoptosis to maintain the germ line in *Drosophila* embryos. *Proceedings of the National Academy of Sciences of the United States of America.* 2007;104(18):7455–60. Epub 2007/04/24. doi: 10.1073/pnas.0610052104. PubMed PMID: 17449640; PubMed Central PMCID: PMC1854842.
6. Hayashi Y, Hayashi M, Kobayashi S. Nanos suppresses somatic cell fate in *Drosophila* germ line. *Proceedings of the National Academy of Sciences of the United States of America.* 2004;101(28):10338–42. PubMed PMID: 15240884.

Dear Prof. Kobayashi,

Thank you for the transfer of your revised manuscript from Review Commons to our editorial offices. I have now received the reports from the three referees that were asked to re-evaluate your study, you will find below. As you will see, the referees support publication of your study in EMBO reports. Nevertheless, referee #3 has suggestions to improve the manuscript I ask you to address in a final revised manuscript. Please discuss the related publication(s) as indicated by the referee. Please also provide a final p-b-p-response addressing this point.

Moreover, the manuscript now also needs formatting according to our journal style. Please carefully review the instructions that follow below.

When submitting your final revised manuscript, we will require:

1) a .docx formatted version of the final manuscript text (including legends for main figures, EV figures and tables), but without the figures included. Figure legends should be compiled at the end of the manuscript text.

We plan to publish your manuscript as Report. For a Scientific Report we require that results and discussion sections are combined in a single chapter called "Results & Discussion". Please do this for your manuscript. For more details please refer to our guide to authors: <http://www.embopress.org/page/journal/14693178/authorguide#researcharticleguide>

2) individual production quality figure files as .eps, .tif, .jpg (one file per figure), of main figures (up to 5 for a report) and EV figures. Please upload these as separate, individual files upon re-submission.

The Expanded View format, which will be displayed in the main HTML of the paper in a collapsible format, has replaced the Supplementary information. You can submit up to 5 images as Expanded View. Please follow the nomenclature Figure EV1, Figure EV2 etc. In this case, please combine and arrange the figures in a way that there will be 5 main figures and up to 5 EV figures.

The figure legend for these should be included in the main manuscript document file in a section called Expanded View Figure Legends after the main Figure Legends section.

For more details, please refer to our guide to authors:
<http://www.embopress.org/page/journal/14693178/authorguide#manuscriptpreparation>

Please consult our guide for figure preparation:
http://wol-prod-cdn.literatumonline.com/pb-assets/embo-site/EMBOPress_Figure_Guidelines_061115-1561436025777.pdf

See also the guidelines for figure legend preparation:
<https://www.embopress.org/page/journal/14693178/authorguide#figureformat>

3) a complete author checklist, which you can download from our author guidelines (<https://www.embopress.org/page/journal/14693178/authorguide>). Please insert page numbers in the checklist to indicate where the requested information can be found in the manuscript. The completed author checklist will also be part of the RPF.

Please also follow our guidelines for the use of living organisms, and the respective reporting guidelines:
<http://www.embopress.org/page/journal/14693178/authorguide#livingorganisms>

4) that primary datasets produced in this study (e.g. RNA-seq, CHIP-seq, structural and array data) are deposited in an appropriate public database. If no primary datasets have been deposited, please also state this in a dedicated section (e.g. 'No primary datasets have been generated and deposited'), see below.

The accession numbers and database should be listed in a formal "Data Availability" section (placed after Materials & Methods) that follows the model below. This is now mandatory (like the COI statement). Please note that the Data Availability Section is

restricted to new primary data that are part of this study. This section is mandatory. As indicated above, if no primary datasets have been deposited, please state this in this section

Data availability

5) We now request the publication of original source data with the aim of making primary data more accessible and transparent to the reader. Our source data coordinator has already contact you to discuss which figure panels we would need source data for. I attach again the source data checklist and a FAQ with instructions.

6) Our journal encourages inclusion of *data citations in the reference list* to directly cite datasets that were re-used and obtained from public databases. Data citations in the article text are distinct from normal bibliographical citations and should directly link to the database records from which the data can be accessed. In the main text, data citations are formatted as follows: "Data ref: Smith et al, 2001" or "Data ref: NCBI Sequence Read Archive PRJNA342805, 2017". In the Reference list, data citations must be labeled with "[DATASET]". A data reference must provide the database name, accession number/identifiers and a resolvable link to the landing page from which the data can be accessed at the end of the reference. Further instructions are available at: <http://www.embopress.org/page/journal/14693178/authorguide#referencesformat>

7) Regarding data quantification and statistics, please make sure that the number "n" for how many independent experiments were performed, their nature (biological versus technical replicates), the bars and error bars (e.g. SEM, SD) and the test used to calculate p-values is indicated in the respective figure legends (also for potential EV and Appendix figures). Please also check that all the p-values are explained in the legend, and that these fit to those shown in the figure. Please provide statistical testing where applicable. Please avoid the phrase 'independent experiment', but clearly state if these were biological or technical replicates. Please also indicate (e.g. with n.s.) if testing was performed, but the differences are not significant. In case n=2, please show the data as separate datapoints without error bars and statistics. See also: <http://www.embopress.org/page/journal/14693178/authorguide#statisticalanalysis>

Please add to each legend (main and EV figures) a 'Data Information' section explaining the statistics used or providing information regarding replicates and scales.

8) Please add scale bars of similar style and thickness to microscopic images, using clearly visible black or white bars (depending on the background). Please place these in the lower right corner of the images themselves. Please do not write on or near the bars in the image but define the size in the respective figure legend.

9) Please also note our reference format:

10) We updated our journal's competing interests policy in January 2022 and request authors to consider both actual and perceived competing interests. Please review the policy <https://www.embopress.org/competing-interests> and update your competing interests if necessary. Please name this section 'Disclosure and Competing Interests Statement' and put it after the Acknowledgements section.

11) We now use CRediT to specify the contributions of each author in the journal submission system. CRediT replaces the author contribution section. Please use the free text box to provide more detailed descriptions and do NOT add an author contributions section to the manuscript text file. See also guide to authors:

<https://www.embopress.org/page/journal/14693178/authorguide#authorshippinguidelines>

12) Please order the manuscript sections like this, using these names:

Title page - Abstract - Keywords - Introduction - Results & Discussion - Methods - Data availability section - Acknowledgements
- Disclosure and Competing Interests Statement - References - Figure legends - Tables
- Expanded View Figure legends

13) Tables S1-S3 are datasets. Please upload these dataset as original excel files with a legend and a title on the first TAB. Please name these files Dataset EV1, Dataset EV2 and Dataset EV3 and change the callouts accordingly.

14) Please enter all the funding information also into our submission system during resubmission and make sure this is complete and similar to the one mentioned in the acknowledgements section of the manuscript text file.

15) Please provide the final abstract written in present tense with not more than 175 words.

16) We would encourage you to use 'Structured Methods', our new Materials and Methods format. According to this format, the Methods section should include a Reagents and Tools Table (listing key reagents, experimental models, software, and relevant equipment and including their sources and relevant identifiers), uploaded as separate file, followed by a Methods and Protocols section in which we encourage the authors to describe their methods using a step-by-step protocol format with bullet points, to facilitate the adoption of the methodologies across labs. More information on how to adhere to this format as well as downloadable templates (.doc or .xls) for the Reagents and Tools Table can be found in our author guidelines (section 'Structured Methods'):

In addition, I would need from you:

- a short, two-sentence summary of the manuscript (not more than 35 words).
- three to four short (!) one sentence bullet points highlighting the key findings of your study.
- a schematic summary figure (synopsis image) in jpeg or tiff format with the exact width of 550 pixels and a height of not more than 400 pixels that can be used as a visual synopsis on our website.

I look forward to seeing a revised version of your manuscript when it is ready. Please let me know if you have questions or comments regarding the revision.

Yours sincerely,

Referee #1 (Referee #2 at Review Commons):

My specific comments on the earlier version of the manuscript have been satisfactorily addressed.

Referee #2 (Referee #1 at Review Commons):

The authors have addressed most of my concerns and I support publication.

Referee #3 (Referee #3 also at Review Commons):

This study reports that repression of somatic gene expression is essential to prevent *Drosophila* germ cells from mixing with somatic cells and dying. The authors identify one soma-expressed gene Miranda whose expression must be suppressed in germ cells to prevent germ/soma mixing.

The authors have addressed some of the technical concerns raised by the reviewers. For example, they have modified the figures to replace bar graphs with plots that show individual biological replicates. The authors were also asked to connect their findings to similar observations reported in 2017 using the *C. elegans* model (Lee et al., 2017). In the revision, the authors have added the Lee et al., 2017 as prior evidence that somatic genes are repressed in PGCs.

Line 81: Expression of the genes required for somatic tissue development are specifically suppressed in primordial germ cells during early embryogenesis (Asaoka et al, 2019; Kojima et al, 2017; Lai et al, 2012; Leatherman et al, 2002; Lee et al, 2017; Martinho et al, 2004; Ohinata et al, 2005; Seydoux et al, 1996; Tomioka et al, 2002; Yamaji et al, 2008).

The authors, however, do not discuss the Lee et al. reference any further, and continue to imply that their findings have no prior precedent in the literature:

Abstract: Our findings uncover a previously unrecognized mechanism whereby somatic gene silencing safeguards the physical boundary between germline and soma, highlighting its potential role in ensuring germline viability during early development.

Line 251: Here, we provide the first report of a molecular mechanism that maintains the spatial separation between pole cells and the soma (Fig. 6).

Line 331: Although no orthologs of *mira* have been reported outside insects (FlyBase, <https://flybase.org/>), other organisms may achieve a similar outcome by repressing different somatic genes through Nanos and global transcriptional silencing mechanisms.

The narrow focus on their own observations is a disservice to readers. Acknowledging that similar findings have been reported using a different animal model would link their observations to a potentially conserved mechanism. I recommend that the authors include a more complete discussion of Lee et al., 2017 to emphasize the similarities, and possible differences (?), between the two model systems. I include below a summary of Lee et al., 2017 and another reference that followed up on that work.

Lee CS, Lu T, Seydoux G. Nanos promotes epigenetic reprogramming of the germline by down-regulation of the THAP transcription factor LIN-15B. *Elife*. 2017 Nov 7;6:e30201. doi: 10.7554/eLife.30201. PMID: 29111977; PMCID: PMC5734877.

Using the *C. elegans* model, Lee et al., 2017 performed transcriptomics on PGCs extracted from Nanos mutants. Similar to the mutants described in the paper by Asaoko et al., Nanos mutants in *C. elegans* cause a subset of PGCs to "mix with soma" outside of the gonad. All germ cells eventually die in Nanos mutants resulting in fully sterile animals (Subramaniam and Seydoux, 1999). Lee et al. 2017 found that Nanos mutant PGCs fail to turn-over a maternally supplied mRNA coding for the THAP transcription factor LIN-15B. They showed that ectopic expression of LIN-15B activates the transcription of somatic genes in PGCs by antagonizing PRC2 activity. They also reported that inactivation of LIN-15B is sufficient to suppress somatic gene transcription in PGCs and restore fertility to Nanos mutants. Based on reports in mammals and *Drosophila* that Nanos is silenced in somatic cells by DRM transcription factors (functionally related to LIN-15B), Lee et al., 2017 proposed that a Nanos-DRM negative feedback loop may be part of an ancient regulatory switch that maintains the germline-soma distinction throughout development.

These findings were confirmed and extended by Cockrum CS, Strome S. Maternal H3K36 and H3K27 HMTs protect germline development via regulation of the transcription factor LIN-15B. *Elife*. 2022 Aug 3;11:e77951. doi: 10.7554/eLife.77951. PMID: 35920536; PMCID: PMC9348848.

It would be interesting for the authors to discuss their observations in the context of the findings in *C. elegans*.

Point-by-point responses to the reviewers' comments

We thank **reviewers 1 and 2** for agreeing with our revisions on the original manuscript.

1) The reviewer 3 stated that “*The authors were also asked to connect their findings to similar observations reported in 2017 using the C. elegans model (Lee et al., 2017). In the revision, the authors have added the Lee et al., 2017 as prior evidence that somatic genes are repressed in PGCs. The authors, however, do not discuss the Lee et al. reference any further, and continue to imply that their findings have no prior precedent in the literature:*

Abstract: *Our findings uncover a previously unrecognized mechanism whereby somatic gene silencing safeguards the physical boundary between germline and soma, highlighting its potential role in ensuring germline viability during early development.*

Line 251: *Here, we provide the first report of a molecular mechanism that maintains the spatial separation between pole cells and the soma.*

Line 331: *Although no orthologs of *mira* have been reported outside insects, other organisms may achieve a similar outcome by repressing different somatic genes through Nanos and global transcriptional silencing mechanisms.*

The narrow focus on their own observations is a disservice to readers. Acknowledging that similar findings have been reported using a different animal model would link their observations to a potentially conserved mechanism. I recommend that the authors include a more complete discussion of Lee et al., 2017 to emphasize the similarities, and possible differences (?), between the two model systems.

It would be interesting for the authors to discuss their observations in the context of the findings in C. elegans.”

Our response

We agree with the comments from reviewer 3. Following her/his suggestion, we have now discussed the observations in *C. elegans* (Lee et al., 2017) as follows.

We used the term “spatial separation (physical segregation) of germline from the soma” in the sense that germline is segregated from the soma forming embryos. For example, primordial germ cells are formed outside the somatic layer and remained there (in the lumen of the midgut primordium) until mid-embryogenesis in insects, such as *Drosophila*, and are transiently located within “extra-embryonic region” outside the soma forming embryo’s body (“embryonic region”) in reptiles, chick, mice, rabbits, and primates.

Our findings in *Drosophila* demonstrate that Nanos/Pgc-dependent double-lock mechanism suppresses the somatic gene, *mira* in the primordial germ cells, thereby maintaining this spatial separation and preventing germline apoptosis. In contrast, the spatial separation of the germline from the soma forming embryo has not been reported in *C. elegans*, although Nanos homologues are active for repressing somatic genes and apoptosis. As seen in the revised manuscript (P14 L327–L332), we speculate that somatic gene repression is a fundamental mechanism that protects germline cells from a harmful somatic environment. This protection may be achieved either through spatial separation from the soma forming embryo’s body, as seen in *Drosophila*, or through an alternative Nanos-dependent mechanism in *C. elegans*. We have added this statement in the latter part of the last paragraph of our revised manuscript.

Abstract: To avoid misunderstanding, we have revised this sentence as follows:
“Our findings uncover a previously unrecognized mechanism whereby somatic gene silencing safeguards the physical boundary between germline and **the somatic cells forming embryo’s body in *Drosophila***, highlighting its potential role in ensuring germline viability during early development.” (P2 L42–L45)

We would like to leave this sentence as it is.

Line 251: We have deleted the paragraph including “Line 251 in the original manuscript”.

Line 331: Regarding “Line 331”, “*other organisms*” did not include the animals, such as *C. elegans*, zebrafish, and *Xenopus*. To avoid misunderstanding, we have added statement about these animals after this sentence (P13 L320–L323). Furthermore, citing the paper (Lee et al., 2017), we have added the notion that Nanos homologues repress LIN-15B transcription factor to prevent germline apoptosis in *C. elegans* (P13 L323–L332), and have also revised the discussion on this in the last paragraph of our revised manuscript (P13 L307–P14 L333).

Dear Prof. Kobayashi,

Thank you for the submission of your further revised manuscript to our editorial offices. After going through your point-by-point response and the revised manuscript, I consider the remaining concerns of referee #3 as adequately addressed.

Before we can proceed with formal acceptance, I have the editorial requests below I ask you to address in a final revised manuscript. Please also provide a final p-b-p-response to the editorial requests.

Editorial requests:

- Please provide the abstract written in present tense throughout.
- Please order the manuscript sections like this, using only these names:
Title page - Abstract - Keywords - Introduction - Results & Discussion - Methods - Data availability section - Acknowledgements (please include here all the funding information) - Disclosure and Competing Interests Statement - References - Figure legends - Expanded View Figure legends
- Please remove the funders listed in the comments box in the submission system (they are already inserted as separate entries).
- Please check again that the number "n" for how many independent experiments were performed, their nature (biological versus technical replicates), the bars and error bars (e.g. SEM, SD) and the test used to calculate p-values is indicated in the respective figure legends (main, EV and Appendix figures). Please also check that all the p-values are explained in the legend, and that these fit to those shown in the figure. Please provide statistical testing where applicable. Please avoid the phrase 'independent experiment' but clearly state if these were biological or technical replicates. Please also indicate (e.g. with n.s.) if testing was performed, but the differences are not significant. In case $n=2$, please show the data as separate datapoints without error bars and statistics. See also:

<https://link.springer.com/journal/44319/submission-guidelines#cms-Figure-and-data-presentation>

If $n < 5$, please show single datapoints for diagrams. Moreover:

- Please note that the legends for figure 2 is not provided in the sequential manner. This needs to be rectified.
- Please note that the box plots need to be defined in terms of minima, maxima, centre, bounds of box and whiskers, and percentile in the legends of figures 2c, e; 3c, d; 4a, c, d, f, h; 5b, c, e, f; EV 1c
- Please note that information related to n is missing in the legends of figures 5b, c "
- Please remove the author list and the affiliations from the title page of the Appendix file. It is sufficient to state here "Appendix for ..." followed by the title of the paper.
- In Figure 4G and 4I the same image is shown (leftmost panels). Please check. If this reuse is intentional, please clearly state this in the figure legend.

In addition, I would need from you uploaded separately:

- a short, two-sentence summary of the manuscript (not more than 35 words).
- two to four short (!) bullet points highlighting the key findings of your study (two lines each). The presently provided bullet points are too long!

I look forward to seeing a new revised version of your manuscript as soon as possible.

Best,

All minor editorial requests have been addressed by the authors.

Prof. Satoru Kobayashi
University of Tsukuba
Life Science Center for Survival Dynamics, Graduate School of Life and Environmental Sciences, Graduate School of Science and Technology
Tsukuba, Ibaraki 305-8577
Japan

Dear Prof. Kobayashi,

I am very pleased to accept your manuscript for publication in the next available issue of EMBO reports. Thank you for your contribution to our journal.

You may qualify for financial assistance for your publication charges - either via a Springer Nature fully open access agreement or an EMBO initiative. Check your eligibility: <https://link.springer.com/journal/44319/how-to-publish-with-us>

Yours sincerely,

>>> Please note that it is EMBO Reports policy for the transcript of the editorial process (containing referee reports and your response letter) to be published as an online supplement to each paper. If you do NOT want this, you will need to inform the Editorial Office via email immediately. More information is available here: <https://link.springer.com/partners/embo-press/editorial-policies#Peer%20review>